# Structure Mediation and Properties of Poly(*l*-lactide)/Poly(*d*-lactide) Blend Fibers

**DOI:** 10.3390/polym10121353

**Published:** 2018-12-06

**Authors:** Bo Yang, Rui Wang, Hui-Ling Ma, Xiaolu Li, Harald Brünig, Zhenfeng Dong, Yue Qi, Xiuqin Zhang

**Affiliations:** 1Beijing Key Laboratory of Clothing Materials R & D and Assessment, Beijing Engineering Research Center of Textile Nanofiber, School of Materials Science & Engineering, Beijing Institute of Fashion Technology, Beijing 100029, China; 15510260661@163.com (B.Y.); clywangrui@bift.edu.cn (R.W.); huiling_1027@163.com (H.-L.M.); m13121937997_1@163.com (X.L.); clydzf@bift.edu.cn (Z.D.); qiyuebift@126.com (Y.Q.); 2Leibniz Institute of Polymer Research Dresden, Hohe Strasse 6, 01069 Dresden, Germany; bruenig@ipfdd.de

**Keywords:** PLLA/PDLA fibers, crystallization, orientation, heat resistance

## Abstract

Poly(*l*-lactic acid) (PLLA) and poly(*d*-lactic acid) (PDLA) blend as-spun fibers (50/50, wt.%) were prepared by melt spinning. Structure mediation under temperature and stress and properties of poly(*l*-lactic acid)/poly(*d*-lactic acid)(PLLA/PDLA) as-spun fibers were investigated by wide-angle X-ray scattering (WAXS) and differential scanning calorimetry (DSC). The results show that highly oriented stereocomplex (SC) crystals can be formed in PLLA/PDLA blend fibers drawn at 60 °C and annealed at 200 °C. However, at drawn temperature of 80 °C, only lower oriented SC crystals can be formed. For PLLA/PDLA blend fibers drawn twice at 60 °C (PLLA/PDLA-60-2), the crystallinity of SC crystals increases with annealing temperature in the range of 200 to 215 °C, while the degree of orientation decreases slightly. When the annealing temperature is 210 °C, the crystallinity and orientation of SC crystals in PLLA/PDLA-60-2 fibers reach 51% and −0.39, respectively. Moreover, PLLA/PDLA-60-2-210 fibers exhibit excellent heat-resistant property even at 200 °C. The results indicate that the oriented PLLA/PDLA blend fibers with high SC crystals content can be regulated in a short time.

## 1. Introduction

Poly(lactic acid) (PLA) has comparable strength with polypropylene, yet it has the advantages of biodegradability [1] and biocompatibility, and so has already been applied as a biomedical material [2,3]. PLA can be made into lightweight fabric which is soft, comfortable, anti-microbial, etc. [4].

However, the poor heat resistance property of PLA materials is detrimental for application in the textile industry [5,6]. Many practical studies have been done by researchers including: Blending with nucleating agents to improve the crystallinity of PLA materials [7,8]; Adding crosslinking agents to form cross-linked structures [9,10]; Adding a certain proportion of PDLA to form SC crystals [11,12]. Among these methods, the SC crystal has attracted great interest in the improvement of PLA heat resistance. The racemic stereocomplex structure was firstly reported by Ikada et al. in the equimolar mixtures of PLLA and PDLA [13]. The melting temperature of SC crystals is about 50 °C higher than that of α crystals in PLLA or PDLA homopolymers (homo-crystals), exhibiting superior heat resistance [14,15]. The enhanced thermal stability of SC crystals provides the possibility to prepare poly(lactic acid) (PLA)-based materials with better dimensional stability at high temperatures [16,17,18]. Therefore, it is industrially promising to prepare PLA fibers with a high content of SC crystals.

Processing conditions (temperature [19,20,21,22] and stress, etc. [23,24,25]) are important factors to regulate the SC crystals in PLA fibers. Tsuji et al. [25] prepared stereocomplex fibers (SC-fibers) by wet and dry spinning processes from mixed solutions of PLLA and PDLA. The fibers prepared by wet spinning processes were difficult to draw further due to the cellular structure on the surface of the fibers, while the dry-spun fibers could be drawn at a high ratio and exhibited good mechanical properties. The ratio of the content of SC to α crystal increased with draw ratio. Takasaki et al. [19] studied the high-speed spinning process of equimolar PLLA/PDLA complexes. The maximum spinning speed reached up to 7.5 km min^−1^. It was found that the content of SC crystals increased with spinning speed. Annealing at temperatures between the melting point of α crystals and SC crystals was also an effective way to increase the SC crystal content in PLLA/PDLA blends by re-crystallization [26,27,28]. However, α crystal would still appear during the cooling process. Furuhashi et al. [20,21] systematically studied the effects of drawing and annealing on crystal structure and mechanical properties of PLLA/PDLA (50/50) complex fibers. The fibers containing pure SC crystals could be obtained by annealing highly oriented or low oriented fibers with some SC crystals. However, the mechanical properties of the stereocomplex fibers were not improved as the SC crystals content increased, which was attributed to the decreasing of orientation during the annealing at the high temperature.

Although the crystal structure and properties of PLLA/PDLA complex have been widely reported [29,30,31], it is still challenging to prepare highly oriented PLA SC-fibers. It is beneficial to anneal fibers at high temperature to form SC crystals. However, the mechanical properties of fibers decrease because of loss of orientation at high temperature [20,21]. Moreover, PLLA/PDLA blend fibers also degrade during the process of annealing at high temperatures, which limits their application. Therefore, how to regulate the oriented PLLA/PDLA blend fibers with high SC crystals content is still a hot topic in recent years.

In our previous work, we have found that drawing at low temperature could maintain the orientation of the fibers when they are annealed at high temperatures [4]. However, the spinning speed of the as-spun fiber was too low, and the fibers need to be annealed at high temperature for a long time to obtain a high content of SC crystals, which is not efficient for practical production. Therefore, in this work, the PLLA/PDLA blend fibers, which were prepared by melt spinning with higher spinning speed, were drawn at 60 °C and 80 °C to form different initial structures. The crystallization behavior of PLLA/PDLA blend fibers with different initial structures was studied. Furthermore, the effect of annealing temperature on the crystallinity and orientation of SC of PLLA/PDLA fibers was investigated. The regulation mechanism of PLLA/PDLA blend fibers with high oriented SC crystals was also examined. The results can provide guidance for the preparation of heat-resistant PLLA/PDLA blend fibers.

## 2. Materials and Methods

### 2.1. Materials

PLLA (optical purity >99%, *M*_w_ = 113 kg/mol, *M*_n_ = 47 kg/mol, polydispersity, DPI = 2.61) was supplied by Zhejiang Haizheng Biomaterial Co., Ltd., Taizhou, China. PDLA (optical purity >98%, *M*_w_ = 140 kg/mol, *M*_n_ = 95 kg/mol, DPI = 1.5) was synthesized by Shandong Daigang Co., Ltd., Jinan, China

### 2.2. Preparation of PLLA/PDLA Blend Fibers

The sample was dried at 80 °C in a vacuum oven (Binder, Germany) for 12 h. Melt spinning of PLLA/PDLA blends was performed at the melt spinning temperature of 245 °C, extrusion speed of 10 mm min^−1^, the capillary diameter of 10 mm and the single-hole spinneret hole diameter of 0.3 mm. PLLA/PDLA as-spun fibers were collected at a take-up speed of 1500 m min^−1^.

### 2.3. Measurements

The melting and crystallization behavior of PLLA/PDLA blends was examined with differential scanning calorimetry(DSC) (TA Instruments, Inc., Newcastle, DE, USA). The instrument was calibrated with indium before measurements. Temperature scans were performed under nitrogen atmosphere. Liquid aluminum crucibles were used. The weight of all samples varied between 5 and 10 mg. The samples were heated to 260 °C and the heating curves were recorded. The heating rate was 10 °C min^−1^.

The as-spun fibers were placed on a Linkam LNP95 stretching hot stage (Linkam Scientific Instruments, Ltd., Tadworth, UK), heated to 60 °C or 80 °C at a rate of 30 °C min^−1^, annealed for 1 min, and then the fibers were stretched to 2 times at the speed of 150 μm s^−1^. After being drawn, the samples were heated to 200 °C annealing for 20 min or 210/215 °C for 1 min, and subsequently cooled to 40 °C. The cooling/heating rate was 5 °C min^−1^. The temperature/draw procedure for in situ wide-angle X-ray scattering (WAXS) is shown in Figure 1. The codes of PLLA/PDLA blend fibers under different conditions are shown in Table 1. In situ WAXS measurements were carried out on X-ray diffractometer(XRD) (Bruker, karlsruhe, ND, USA). The wavelength of the radiation source was 0.154 nm. The scattering patterns were collected by a MAR CCD (MAR-USA) detector, which had a resolution of 2048 × 2048 pixels (pixel size = 79 × 79 mm^2^). The sample-to-detector distance was 85.6 mm and the image acquisition time was 60 s. The samples were tested for every 10 °C during heating and cooling process. All the images were corrected for background scattering, air scattering and sample absorption.

The one-dimensional (1D) diffraction intensity for each 2θ was obtained by integration over the azimuthal range (60°–120°) of the 2D diffraction images. To estimate the fraction of different phases, the WAXS intensity profile was deconvolved into several Gaussian profiles to represent all of the visible scattering peaks and the amorphous halo. The relative fraction of different phases (X) of the samples was calculated by the following equations:
(1)$$X_{\alpha }=\frac{I_{\alpha }}{I_{\alpha }+I_{sc}+I_{amor}}\;,$$
(2)$$X_{sc}=\frac{I_{sc}}{I_{\alpha }+I_{sc}+I_{amor}}\;,$$
(3)$$X=X_{\alpha }+X_{sc}\;,$$
where *I_α_*, *I_sc_*, and *I_amor_* stand for the integral intensities of the peaks of α crystals, SC crystals and amorphous phase, respectively [32].

The orientation factor of the (110)/(200) crystal plane is calculated according to the Hermans formula [33], where Φ is the angle between the normal direction of the crystal plane and the fiber axis.
(4)$$<\cos^{2}\Phi >=\frac{\int_{0}^{\frac{\pi }{2}}I\left(\Phi \right)\cos^{2}\Phi \sin\Phi d\Phi }{\int_{0}^{\frac{\pi }{2}}I\left(\Phi \right)\sin\Phi d\Phi }\;,$$
(5)$$f_{\left(200\right)/\left(100\right)}=\frac{3\left({<\cos}^{2}\Phi >\right)-1}{2}\;,$$

The fibers were placed in an oil bath with different temperatures for 1 min. The original length (*L*_0_) and the final length (*L*_1_) were measured and the shrinkage S was calculated by the following equation:(6)$$S\left(\%\right)=\frac{L_{0}-L_{1}}{L_{0}}\times 100\%,$$

## 3. Results

### 3.1. The Initial Structure of PLLA/PDLA As-Spun Fibers

WAXS intensity profile of PLLA/PDLA as-spun fibers is shown in Figure 2a. No reflection peak is observed in the WAXS profile, and only an amorphous halo appears, indicating amorphous phase in the PLLA/PDLA as-spun fibers. As shown in Figure 2b, the endothermic peak at 59.92 °C is glass transition and thermal enthalpy relaxation. The sharp exothermal peak at 90.64 °C is assigned to cold crystallization. Zhang et al. [34] reported that α’ crystals would generate below 120 °C, which was less stable than α crystals. The small exothermic peak at 157.32 °C corresponds to the transition from α’ to α crystals [34,35]. The two endothermic peaks at 173.33 °C and 224.20 °C are assigned to the melting peaks of α crystals and SC crystals, respectively. From these results, it can be concluded that the prepared as-spun fibers are amorphous.

### 3.2. Crystal Formation of Fibers with Different Initial Structures during Heating and Cooling Process

It is known that the initial structures of PLLA/PDLA blend fibers influence the structure evolutions of fibers upon heating and cooling. To acquire the different initial structures, the stretching temperatures of 60 and 80 °C were selected. The crystallization variation of PLLA/PDLA blend fibers with different initial structure during heating and cooling process is shown in Figure 3. The unstretched as-spun fibers, PLLA/PDLA-60 and PLLA/PDLA-80, exhibited similar behavior during the heat treatment. The initial structure was almost amorphous. During heating, α crystals with a low degree of orientation appeared gradually. SC crystals, with a negligible degree of orientation, appeared at 200 °C. Upon cooling, isotropic α crystals formed again. These results show that the orientation of fiber is lost although the amount of SC crystals increases in the fibers.

PLLA/PDLA-60-2 displays a diffuse halo, indicating the amorphous structure PLLA/PDLA fibers. When the temperature reached 100 °C, the reflections of (110)/(200) and (203) corresponding to the α crystals was observed at 2θ = 16.1° and 18.3° in PLLA/PDLA-60-2 fibers [36], suggesting the formation of α crystals. A strong diffraction spot reflection at 2θ = 11.6 ° was observed at 200 °C, corresponding to the (110) of SC crystals, indicating that the SC crystals in PLLA/PDLA-60-2 fibers are highly oriented [37]. Then, the fibers were annealed at 200 °C for 20 min and the oriented α crystals melted. It is noteworthy that strong diffraction spots corresponding to SC crystals still existed in the pattern, demonstrating that the high orientation can be maintained at 200 °C in PLLA/PDLA-60-2 fibers. When PLLA/PDLA-60-2 fibers were cooled to 150 °C, the reflections of (110)/(200) assigned to α crystals appeared at 16.1°, and formed α crystals that were also highly oriented.

For PLLA/PDLA-80-2, diffraction of (110)/(200)α is observed at 16.1 °. The PLA segments can move to form oriented α crystals because the drawing temperature (80 °C) is higher than *T*_g_ of PLA. When the fibers are heated to 100 and 150 °C, only α crystals are formed in PLLA/PDLA blend fibers. With temperature up to 200 °C for 20 min, diffraction arc of SC crystals is observed at 11.6°. However, SC crystals reflections are obtuse, suggesting that the formed SC crystals are less oriented. The contents of SC crystals, α crystals and orientation factor of the PLLA/PDLA fibers are listed in Table 2 and Table 3. The result shows that PLLA/PDLA-60-2 has higher SC crystals content and degree of orientation, which is attributed to the higher orientation of PLA segments in PLLA/PDLA-60-2 at low drawing temperature. When the fibers are annealed at 200 °C, the higher PLA oriented segments improve crystallization rate of SC crystals, thus suppress the relaxation of oriented PLA segments and form highly oriented SC crystals. The content of SC crystals in PLLA/PDLA-60-2 fibers reaches more than 30% (Table 2), and the value of ƒ(110)_sc_ reaches −0.40 (Table 3). These results suggest that oriented SC crystals with higher content can be formed in the PLLA/PDLA-60-2 fibers in this unique treatment process.

### 3.3. Effect of Annealing Temperature on Crystallization of PLLA/PDLA Blend Fibers

The annealing temperature is a key factor to regulate the formation of SC crystals in the PLLA/PDLA fibers. In our previous study [38], the PLLA/PDLA (50/50) film were annealed at 185–225 °C for 30 min. The results suggest that the fraction of SC crystals increased from 16% to 35% with increasing annealing temperature. It is believed that SC crystals formed at the interface acting as a barrier layer so as to hinder the further diffusion of the chains. When the annealing temperature increases to melt the “SC crystal barrier wall”, the enhanced chain diffusion between PLLA and PDLA enrichment region will improve the formation of SC crystals. Therefore, the effect of annealing temperature on the crystallization behavior of PLLA/PDLA-60-2 fibers needs further exploration to obtain a higher content of SC crystals with orientation.

The 2D WAXS patterns of the PLLA/PDLA-60-2 fibers upon heating are shown in Figure 4. When the PLLA/PDLA-60-2 fibers are heated to 220 °C, SC crystals of PLLA/PDLA-60-2 are partially melted and the orientation decreases because the melting temperature of SC crystals is lower than 220 °C. Further increasing temperature to 225 °C, the crystals are completely melted. In order to maintain the crystal orientation of PLLA/PDLA-60-2 fibers during annealing and cooling process, the annealing temperatures of 210 and 215 °C are selected.

The effect of annealing temperature on the crystallization of PLLA/PDLA-60-2 fibers is shown in Figure 5. When PLLA/PDLA-60-2 fibers are heated to 210 or 215 °C for 1 min, the diffraction of SC crystals becomes sharper and stronger, which indicates the higher crystallinity of SC crystals. When the fibers are cooled down to the room temperature slowly, the diffraction of α crystals is very weak. The oriented SC crystals are the main crystal structure in the fibers. The fractions of α crystals and SC crystals in the PLLA/PDLA-60-2 fibers treated with different processes are estimated by the deconvolution of the WAXS intensity profiles (Figure 6). During the cooling process, the content of SC crystals in PLLA/PDLA-60-2 fibers increases when the temperature is higher than 180 °C. The maximum content of SC crystals in both PLLA/PDLA-60-2-210 and PLLA/PDLA-60-2-215 fibers is about 50% (Table 4). The content of α crystals is very low, indicating that high content of SC crystals in the fibers suppresses the formation of α crystal [39,40]. As shown in Table 4, with the annealing temperature of 200 and 215 °C, the content of SC crystals increases from 32% to 51%, and the value of ƒ(110)_sc_ decreases from −0.40 to −0.36. Furthermore, the melting temperature of SC crystals reaches up to 231.70 °C with annealing temperature up to 215 °C (Figure 7). According to the results of crystallinity and orientation of SC crystals, the annealing temperature of 210 °C is optimal.

### 3.4. Heat-Resistant Properties of PLLA/PDLA Blend Fibers

The heat resistance property of PLA fibers was investigated by placing PLLA-130 (annealing at 130 °C for 30 min) and PLLA/PDLA-60-2-210 (annealing at 210 °C for 1 min) fibers into the oil bath. The shrinking results of PLLA-130 and PLLA/PDLA-60-2-210 fibers at different temperatures are listed in Table 5 and Figure 8. The shrinkage percentage of PLLA-130 fibers reaches 48% at 150 °C. When the temperature is 200 °C, PLLA-130 fibers are molten, while PLLA/PDLA-60-2-210 fibers maintain the original length without significant shrinkage. Thus, high content of SC crystals endows the fibers with better heat resistance. When the temperature reaches 210 °C, the PLLA/PDLA-60-2-210 fibers also begin to shrink because this temperature is already close to the melting temperature of SC crystals. PLLA/PDLA-60-2-210 fibers are completely molten at 220 °C.

## 4. Conclusions

In this work, PLLA/PDLA as-spun fibers were prepared by melting spinning. The obtained fibers were drawn up to two times at 60 °C and 80 °C, respectively. The content of SC crystals in PLLA/PDLA-60-2 fibers was 32% when annealing at 200 °C for 20 min, and the degree of orientation was up to −0.40, higher than that of PLLA/PDLA-80-2 fibers. When the annealing temperature increased from 200 to 215 °C, the content of SC crystals of PLLA/PDLA-60-2 fiber increased from 32% to 51%, and the value of *f*(110)_sc_ decreased from −0.40 to −0.36. According to the results of crystallinity and orientation of SC crystals, the annealing temperature of 210 °C was the best, and the content of SC crystals in PLLA/PDLA-60-2 fibers was 51%, and the value of *f*(110)_sc_ orientation was −0.39. Moreover, the PLLA/PDLA-60-2-210 fibers exhibited excellent heat resistance and the shrinkage ratio was 27% at 210 °C.

## Reference

## Figures and Tables

**Figure 1 polymers-10-01353-f001:**
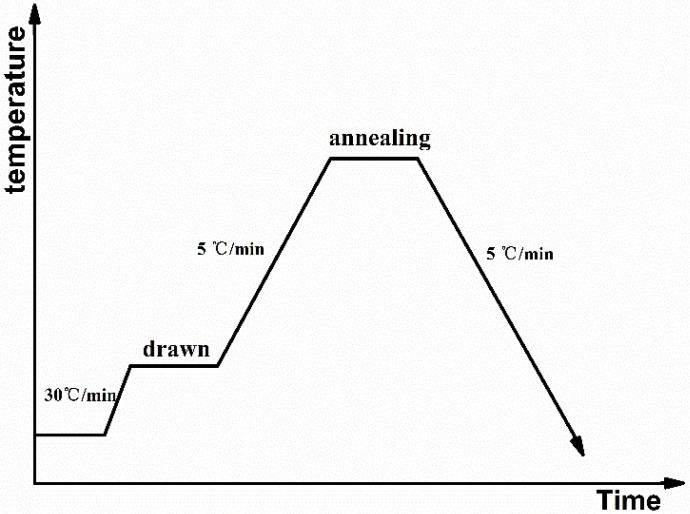
Temperature/draw procedure for in situ WAXS.

**Figure 2 polymers-10-01353-f002:**
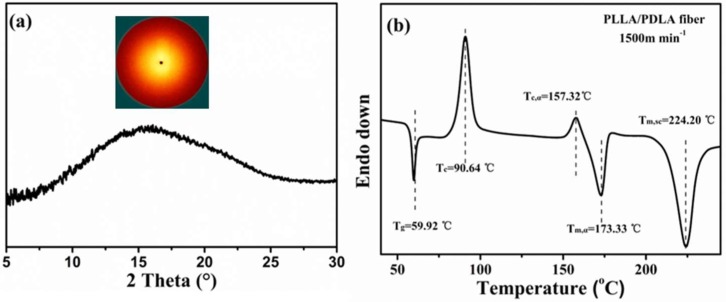
(**a**) WAXS intensity curve and (**b**) DSC heating curve of PLLA/PDLA blend as-spun fiber.

**Figure 3 polymers-10-01353-f003:**
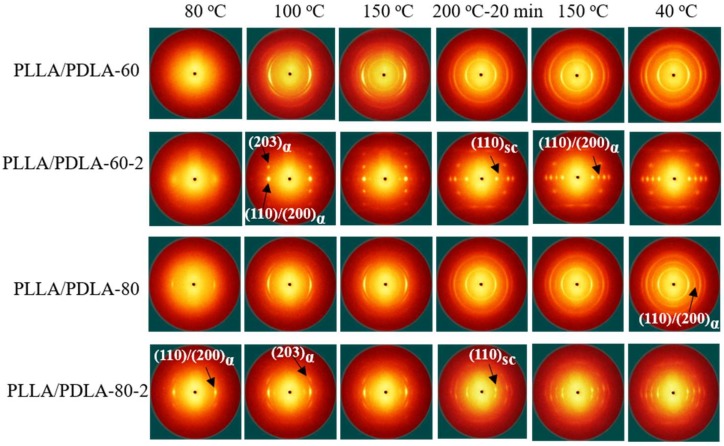
Selected 2D WAXS patterns of PLLA/PDLA fibers during heating and cooling. The stretching direction was vertical.

**Figure 4 polymers-10-01353-f004:**
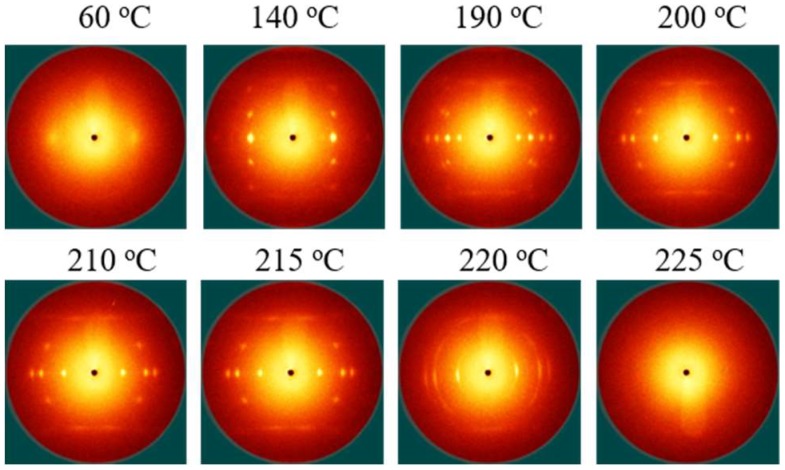
Selected 2D WAXS patterns of PLLA/PDLA-60-2 fiber during the heating process.

**Figure 5 polymers-10-01353-f005:**
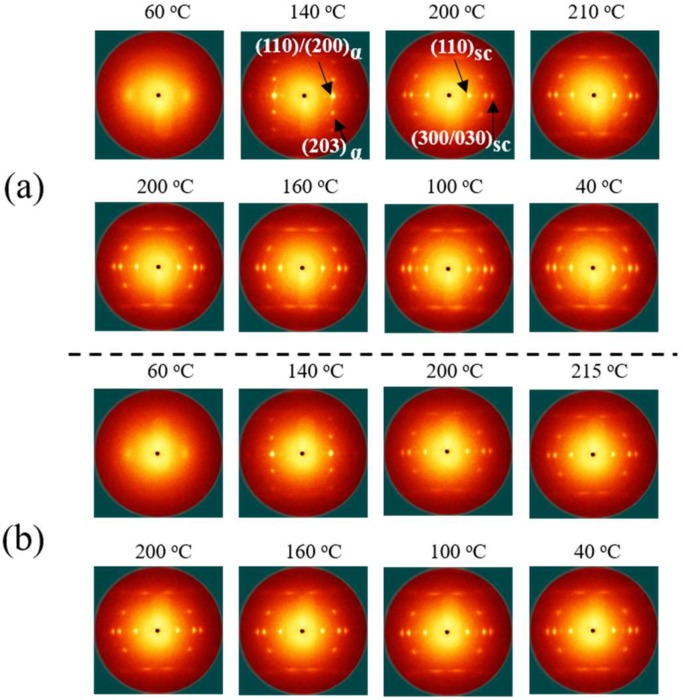
Selected 2D WAXS patterns of PLLA/PDLA-60-2 fibers during heating to 210 °C (**a**) and 215 °C (**b**), and the following cooling process.

**Figure 6 polymers-10-01353-f006:**
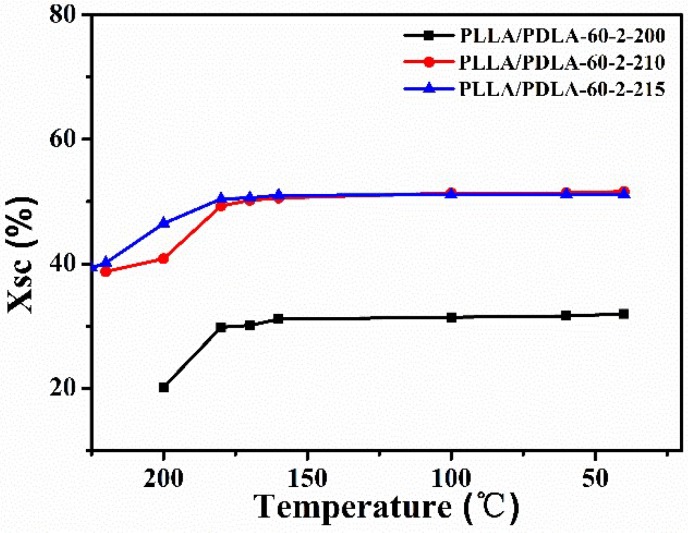
Crystallinity calculated from the WAXS results shown in Figure 3 and Figure 5 for PLLA/PDLA-60-2 fibers after cooled from different temperatures.

**Figure 7 polymers-10-01353-f007:**
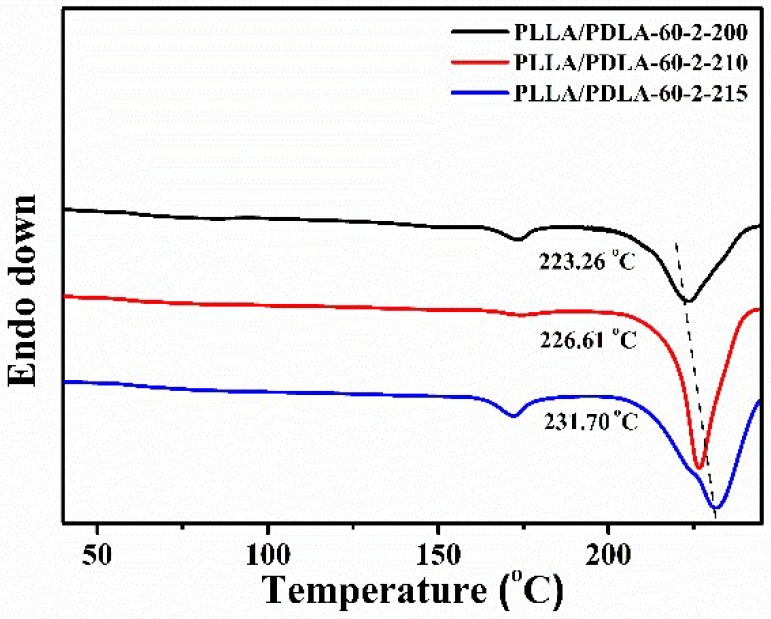
DSC heating curves of PLLA/PDLA-60-2 fibers annealed at various temperatures.

**Figure 8 polymers-10-01353-f008:**
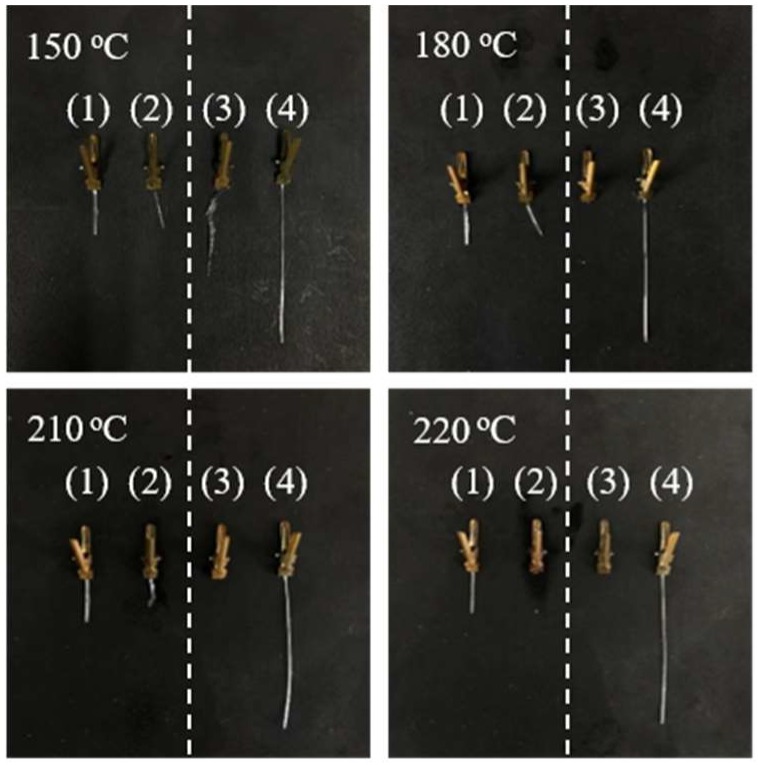
Appearance of PLA fibers. (**1**) PLLA/PDLA-60-2-210 fibers (original length 1.1 cm), (**2**) PLLA/PDLA-60-2-210 fibers (heated at different temperatures), (**3**) PLLA-130 fibers (heated at different temperatures), (**4**) PLLA-130 fibers (original length 4.3 cm).

**Table 1 polymers-10-01353-t001:** Code of PLLA/PDLA blend fibers.

Conditions	Code
60 °C undrawn	PLLA/PDLA-60
60 °C drawn up to 2 times	PLLA/PDLA-60-2
80 °C undrawn	PLLA/PDLA-80
80 °C drawn up to 2 times	PLLA/PDLA-80-2
60 °C drawn up to 2 times, annealed at 210 °C	PLLA/PDLA-60-2-210
60 °C drawn up to 2 times, annealed at 215 °C	PLLA/PDLA-60-2-215

**Table 2 polymers-10-01353-t002:** The phase contents of PLLA/PDLA fibers during annealing at 200 °C and cooling to room temperature.

Sample	*X* _0min-A_	*X* _10min-A_	*X* _20min-A_	*X* _α_	*X* _sc_
PLLA/PDLA-60	21	24	24	1.4	25
PLLA/PDLA-60-2	20	21	20	4.9	32
PLLA/PDLA-80	21	24	23	4.1	29
PLLA/PDLA-80-2	19	22	21	5.2	29

**Table 3 polymers-10-01353-t003:** The orientation factor of PLLA/PDLA fibers.

Sample	*f* _(110)/(200)α_	*f* _(110)sc_
PLLA/PDLA-60	/	−0.15
PLLA/PDLA-60-2	−0.25	−0.40
PLLA/PDLA-80	/	−0.13
PLLA/PDLA-80-2	−0.13	−0.21

**Table 4 polymers-10-01353-t004:** The phase content and orientation factor of PLLA/PDLA-60-2 fibers cooling from various temperatures.

T (°C)	X_α_ (%)	X_sc_ (%)	*f* _(110)sc_
200	7.4	32	−0.40
210	2.0	51	−0.39
215	0.93	51	−0.36

**Table 5 polymers-10-01353-t005:** Shrinking percentages of PLLA-130 and PLLA/PDLA-60-2-210 fibers at different temperatures.

Temperature (°C)	PLLA-130 (%)	PLLA/PDLA-60-2-210 (%)
130	2.3 ± 0.4	0
140	9.3 ± 0.4	0
150	48 ± 0.6	0
160	71 ± 1.2	0
170	91 ± 1.2	0
180	melt	0
190	melt	0
200	melt	0
210	melt	27 ± 0.5
220	melt	melt

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
