# Peer review of "Structure Mediation and Properties of Poly(l-lactide)/Poly(d-lactide) Blend Fibers"

_polymers, 2018, doi:10.3390/polym10121353_

Reviewer 1 Report

I have reviewed your intriguing manuscript, which studies the effect of  Poly (L-lactic acid) (PLLA) and poly (D-lactic acid) (PDLA) blend as-spun fibers (50/50, 12 wt%) prepared by melt spinning.

You provide a detailed series of investigations, encompassing wide-angle X-ray scattering (WAXS) 14 and differential scanning calorimetry (DSC).

The work has scientific merits, and is generally well written. The experiments are well designed, and most conclusions are supported by the data. However, a range of shortcomings attracted my attention, which need to be addressed to improve your submission:

1.        Figure 1 is quite illegible
2.        please explain all the abbreviations in the manuscript
3.        I do not agree with Authors:   Moreover, 21 PLLA/PDLA-60-2-210 fibers exhibit excellent heat resistant property even at 210 °C
4.         Line 57 to 63 please add references
5.        why PLA was dried under such conditions ?: The sample was dried at 80 °C in a vacuum oven (Binder, Germany) for 12 h.
6.        which melting-pot were used in the DSC analysis ?
7.        there is a lack of many details in the research methodology

I reconsider the manuscript after major revision.

Author Response

Dear Editor,

Thank you very much for giving us a chance to revise our manuscript (Manuscript ID: polymers-397502). Also, we are grateful for the reviewers’ constructive comments and criticisms, which could help to improve the quality of the paper.

According to the reviews’ comments, we have carefully revised the manuscript. A point-to-point response to the comments was made. Corresponding revisions were marked in a different color in the manuscript. Some new data was provided in the supporting information.

Now, we submitted it to you for your further consideration.

Best regards.

Xiuqin Zhang and co-authors

Response to Reviewer 1 Comments

Reviewer #1

The work has scientific merits, and is generally well written. The experiments are well designed, and most conclusions are supported by the data. However, a range of shortcomings attracted my attention, which need to be addressed to improve your submission.

Point 1: Figure 1 is quite illegible.

Response 1: Thank you for your kind suggestion. We have redrawn Figure 1.

Point 2: Please explain all the abbreviations in the manuscript.

Response 2: All the explanations of abbreviations were added in the revised manuscript. (See Page 3, Table1)

Point 3: I do not agree with Authors: Moreover, PLLA/PDLA-60-2-210 fibers exhibit excellent heat resistant property even at 210 °C.

Response 3: Thank you for your insightful suggestion. We have carefully checked the phenomena and experimental data according to the reviewer’s suggestion. The results suggested that PLLA/PDLA-60-2-210 fibers exhibited excellent heat resistant property even at 200 °C. The corresponding description was revised in Line 261-266, Page10.

Point 4: Line 57 to 63 please add references.

Response 4: Thank you for the suggestion. Five references were cited as ref. 20, ref. 21, ref. 29 ref. 30 and ref. 31.(Page 2, Line 60 and Line 62)

Point 5: Why PLA was dried under such conditions? The sample was dried at 80 °C in a vacuum oven (Binder, Germany) for 12 h.

Response 5: PLA could be degraded by the hydrolysis of ester bonds in an active and humid environment, resulting in significant drop in molecular weight, which seriously affected the quality of fibers. According to Furuhashi’s work, PLLA and PDLA could be dried in a vacuum oven at 80 oC for 1 h and then at 130 oC for 6 h prior to the melt spinning. (Polymer 2006, 47, 5965-5972). To avoid degradation at high temperaturesWe adopt a method of extending the drying time at 80 ℃. It has been proved that the drying condition is suitable through spinning and melt processing. (European Polymer Journal, 2016, 82, 46-56). So we adopt the appropriate temperature and extend time.

Point 6: Which melting-pot were used in the DSC analysis ?

Response 6: Liquid aluminum crucibles were used in this work. We have added it in the revised manuscript.

Point 7: There is a lack of many details in the research methodology.

Response 7: Thank you for your suggestion. We have revised and added some details in the “measurements part”, on Page 3, Line 93-104.

Reviewer 2 Report

The article ‘Structure mediation and properties of poly(L-lactide) 2 /poly(D-lactide) blend fibers’ describes the physical properties of polymer bled fibers, with a focus on the crystallinity development and stereocomplex composition. Overall the manuscript is well presented, with good quality figures. The heat resistance of these fibers is an additional highlight.

The manuscript can be improved, considering the points listed below. This does not require further experiments, and can be considered as a ‘minor revision’.

1.       The abstract lacks a concluding sentence, summarizing the key results or highlights of the study.

2.       The introduction section should address the advantages of stereocomplex crystalline states. There are several interesting applications in literature, for instance, see:

Li, Wei, et al. "The origin of memory effect in stereocomplex poly (lactic acid) crystallization from melt state." European Polymer Journal 89 (2017): 241-248.

Barbosa, P., et al. "Piezoelectric poly (lactide) stereocomplexes with a cholinium organic ionic plastic crystal." Journal of Materials Chemistry C 5.46 (2017): 12134-12142.

Xie, Yan, et al. "High-performance porous polylactide stereocomplex crystallite scaffolds prepared by solution blending and salt leaching." Materials Science and Engineering: C 90 (2018): 602-609.

3.       How many replicate experiments were performed? Especially for the shrinkage studies, provide error values.

4.       Does the values of 51% for the degree of crystallinity represent the maximum, depending on the properties of the polymer? Or is there potential to increase this value?

A related issue is the optimal annealing temperature of 210C. This temperature values is possibly specific for the properties of the applied polymers. Hence it is importance to state its dependence on polymer physico-chemistry.

5.       Which crucibles were used for DSC?

6.       Please check carefully for grammatical errors:

Line 72: oriented

Line 119: the a crystal….does it mean, the α crystal? Or is it a grammatical error?

Line 126: L0, L1. Please use the suffixes, as stated in equation (6)

Author Response

Dear Editor,

Thank you very much for giving us a chance to revise our manuscript (Manuscript ID: polymers-397502). Also, we are grateful for the reviewers’ constructive comments and criticisms, which could help to improve the quality of the paper.

According to the reviews’ comments, we have carefully revised the manuscript. A point-to-point response to the comments was made. Corresponding revisions were marked in a different color in the manuscript. Some new data was provided in the supporting information.

Now, we submitted it to you for your further consideration.

 Best regards.

Response to Reviewer 2     Comments

Reviewer #2

The article ‘Structure mediation and properties of poly(L-lactide) 2 /poly(D-lactide) blend fibers’ describes the physical properties of polymer bled fibers, with a focus on the crystallinity development and stereocomplex composition. Overall the manuscript is well presented, with good quality figures. The heat resistance of these fibers is an additional highlight.

Point 1: The abstract lacks a concluding sentence, summarizing the key results or highlights of the study.

Response 1: We have revised the abstracts and summarized the key results according to your kind suggestion. (Page 1, Line 21-24)

Point 2: The introduction section should address the advantages of stereocomplex crystalline states. There are several interesting applications in literature, for instance, see:

Li, Wei, et al. "The origin of memory effect in stereocomplex poly (lactic acid) crystallization from melt state." European Polymer Journal 89 (2017): 241-248.

Barbosa, P., et al. "Piezoelectric poly (lactide) stereocomplexes with a cholinium organic ionic plastic crystal." Journal of Materials Chemistry C 5.46 (2017): 12134-12142.

Xie, Yan, et al. "High-performance porous polylactide stereocomplex crystallite scaffolds prepared by solution blending and salt leaching." Materials Science and Engineering: C 90 (2018): 602-609.

Response 2: Thank you for the suggestion. The suggested references were added as ref. 29, 30 and 31. (Page 2, Line 60)

Point 3: How many replicate experiments were performed? Especially for the shrinkage studies, provide error values.

Response 3: The replicate experiments were carried out at least five times, and the average value can be obtained. We have also revised the results in Table 5 on Page 9.

Point 4: Does the values of 51% for the degree of crystallinity represent the maximum, depending on the properties of the polymer? Or is there potential to increase this value?

A related issue is the optimal annealing temperature of 210 ℃. This temperature values is possibly specific for the properties of the applied polymers. Hence it is importance to state its dependence on polymer physico-chemistry.

Response 4: According to our previous work, Xiong et al added nucleating agent TMP-5(4 wt%) into PLLA/PDLA (50/50), the content of stereocomplex crystals (SC) reaches to 40%. And Yin et al reported that PLLA/PDLA(50/50) blends were annealed at 225 ℃ for 30 min, The maximum content of crystal was 55%. (European Polymer Journal, 2016, 82, 46-56). So we believe that the values of 51% for the degree of crystallinity represent the maximum. On the other hand, Yin et al found that the PLLA/PDLA (50/50) film were annealed at 185-225 °C for 30 min, the results suggested that the fraction of SC crystals increased from 16% to 35% with increasing annealing temperature. It is believed that SC crystals formed at the interface acting as a barrier layer so as to hinder the further diffusion of the chains. When the annealing temperature increases to melt the "SC crystal barrier wall", the enhanced chain diffusion between PLLA and PDLA enrichment region will improve the formation of SC crystals. In this article, the orientation would decrease with the increasing temperature, which was shown in Table 4. Therefore, annealing temperature of 210 ℃ was optimal, the PLLA/PDLA fibers had high content of SC crystals with highly orientation.

Point 5: Which crucibles were used for DSC?

Response 5: Liquid aluminum crucibles were used in this work.

Point 6: Please check carefully for grammatical errors:

Line 72: oriented

Line 119: the a crystal….does it mean, the α crystal? Or is it a grammatical error?

Line 126: L0, L1. Please use the suffixes, as stated in equation (6)

Response 6: Thanks for the reviewer’s kind suggestions of our manuscript. These errors have been corrected in the revised manuscript. (Line 74, Line 118 and Line 125). Meanwhile, the whole manuscript has been carefully checked to avoid some grammatical and spelling errors.

Round  2

Reviewer 1 Report

I accept the manuscript  in present form